# Infertile men with semen parameters above WHO reference limits at first assessment may deserve a second semen analysis: Challenging the guidelines in the real-life scenario

**Luca Boeri**[1], **Edoardo Pozzi**[2,3], **Paolo Capogrosso**[4], **Giuseppe Fallara**[2,3], **Federico Belladelli**[2,3], **Luigi Candela**[2,3], **Nicolò Schifano**[2,3], **Christian Corsini**[2,3], **Walter Cazzaniga**[2,3], **Daniele Cignoli**[2,3], **Eugenio Ventimiglia**[2], **Marina Pontillo**[5], **Massimo Alfano**[2], **Francesco Montorsi**[2,3], **Andrea Salonia**[2,3]*

**1** Department of Urology, Foundation IRCCS Ca' Granda–Ospedale Maggiore Policlinico, Milan, Italy,
**2** Division of Experimental Oncology/Unit of Urology; URI; IRCCS Ospedale San Raffaele, Milan, Italy,
**3** Vita-Salute San Raffaele University, Milan, Italy, **4** Department of Urology and Andrology, Ospedale di Circolo and Macchi Foundation, Varese, Italy, **5** Laboratory Medicine Service, IRCCS Ospedale San Raffaele, Milan, Italy

* salonia.andrea@hsr.it

**Data Availability Statement:** All relevant data are within the manuscript and its Supporting Information files.

## Abstract

### Objectives

To investigate which infertile men with semen parameters above WHO reference limits at first semen analysis deserve a second semen test.

### Materials and methods

Data from 1358 consecutive infertile men were analysed. Patients underwent two consecutive semen analyses at the same laboratory. Descriptive statistics and logistic regression models tested the association between clinical variables and semen parameters. A new predicting model was identified through logistic regression analysis exploring potential predictors of semen parameters below WHO reference limits after a previously normal one. Diagnostic accuracy of the new model was compared with AUA/ASRM and EAU guidelines. Decision curve analyses (DCA) tested their clinical benefit.

### Results

Of 1358, 212 (15.6%) infertile men had semen parameters above WHO reference limits at first analysis. Of 212, 87 (41.0%) had a second semen analysis with results above WHO reference limits. Men with sperm parameters below reference limits at second analysis had higher FSH values, but lower testicular volume (TV) (all p<0.01) compared to men with a second semen analysis above WHO limits. At multivariable logistic regression analysis, lower TV (OR 0.9, p = 0.03), higher FSH (OR 1.2, p<0.01), and lower total sperm count (OR 0.9, p<0.01) were associated with second semen analyses below WHO limits. DCA showed the superior net benefit of using the new model, compared to both AUA/ASRM and EAU

**Funding:** The author(s) received no specific funding for this work.

**Competing interests:** The authors have declared that no competing interests exist.

guidelines to identify those men with a second semen sample below WHO limits after a previously normal one.

## Conclusions

Approximately 60% of infertile men with a first semen analysis above WHO limits have a second analysis with results below limits. The newly identified risk model might be useful to select infertile men with initial semen results above WHO limits who deserve a second semen analysis.

## Introduction

Semen analysis is considered the cardinal point throughout the investigation of men presenting for couple's infertility, along with a comprehensive medical and reproductive history, physical examination and hormonal investigation [1, 2]. Nonetheless, the individual semen parameter provides only partial information of the actual fertility potential. In fact, even having strictly normal sperm parameters according to World Health Organization (WHO) reference values [3, 4] per se does not reliably account for fertility [5]. In this context, a recent case-control study showed that sperm parameters above WHO limits were found in approximately 12% of infertile men and in only 41% of age-matched fertile controls [5]. Moreover, between 20% and 30% of men are infertile despite having semen analysis above limits, normal medical history and normal physical examination, thus configuring the condition of unexplained male infertility [1, 6]. As a whole, semen parameters are highly variable biological measures and may vary substantially from ejaculate to ejaculate. Therefore, the American Urological Association/American Society for Reproductive Medicine (AUA/ASRM) guidelines and WHO references criteria recommend analysing at least two consecutive samples in every infertile man [2, 7]; conversely, the European Association of Urology (EAU) Guidelines on Sexual and Reproductive health suggests that a single test is sufficient in case of normal semen analysis according to WHO reference criteria, whereas a second assessment should be taken into consideration in case of sperm abnormalities [1, 8]. According to physiological spermatogenesis, a second analysis is considered reliable after a 3-month time frame [9]; thereof, if not adequately collected, a second semen analysis could potentially lead to unnecessary costs and delays over the treatment work-up of infertility, and previous studies have investigated the need to collect more than one semen sample in clinical practice [10–13]. As a whole, a number of studies did not find significant differences among semen parameters obtained on consecutive analyses both in fertile and infertile men [10, 11, 13]. Conversely, other reports showed greater intraindividual variations in infertile men compared to sperm donors [12, 14]. Therefore, the overall clinical benefit of performing one versus two consecutive semen analyses in infertile men is still a matter of debate. This is even more true in infertile men with semen parameters above WHO limits at first semen sample, where current scientific guidelines are discordant in suggesting the need for a confirmatory second test [1, 2]. For instance, in their cohort of 2,566 infertile men, Blickenstorfer et al. showed that in cases with semen sample above WHO limits at first assessment, 27% of the second samples were below limits [10]. Thus, a proper identification of those men with semen samples above WHO limits who would truly benefit from a second investigation in the real-life setting is needed.

Thereof, we aimed to investigate: i) the rate of and the predictors of semen analyses below WHO limits at second investigation after a first one above limits in a homogeneous cohort of non-Finnish, white-European men seeking medical attention for primary couple's infertility;

and ii) to build a predictive model for second semen analysis below WHO limits that could be used to identify infertile men that deserve a confirmatory test over the diagnostic work-up.

## Materials and methods

Data from a cohort of 1562 consecutive white-European men assessed at a single academic centre for couple's infertility (non-interracial infertile couples only) between September 2012 and September 2020 were analysed. According to the WHO criteria, infertility was defined as not conceiving a pregnancy after at least 12 months of unprotected intercourses regardless of whether or not a pregnancy ultimately occurs [15]. Patients were only enrolled if they were ≥18 and ≤55 years old and had either pure male factor infertility (MFI) or mixed factor infertility. MFI was defined after a comprehensive diagnostic evaluation of all the female partners by expert Gynaecologists, which included a detailed medical, reproductive and family history as well as a general and gynaecological physical examination. Furthermore, the ovulatory status, ovarian reserve testing, the structure and patency of the female reproductive tract were requested in all cases.

All participants were homogenously assessed by the same expert academic urologist (A.S.), with a thorough medical history and a complete physical examination. Health-significant comorbidities were scored with the Charlson Comorbidity Index (CCI), coded using the International Classification of Diseases, 9[th] revision [16, 17]. Likewise, weight and height were measured, calculating body mass index (BMI) for each participant [18]. Testes volume (TV) was assessed in all cases using Prader's orchidometer estimation [19]; for the specific purpose of this study, we calculated the mean value between the two sides. Varicocele was also clinically assessed in every patient [20]. Venous blood samples were drawn from each patient between 7 AM and 11 AM after an overnight fast. Follicle-stimulating hormone (FSH), luteinizing hormone (LH), total testosterone (tT), prolactin, thyroid-stimulating hormone (TSH) and sex hormone-binding globulin (SHBG) levels were measured for every individual. Sperm DNA fragmentation (SDF) index, was measured by sperm chromatin structure assay (SCSA) [21]. According to our internal diagnostic protocol, chromosomal analysis and genetic testing were performed in every infertile man (i.e., karyotype analysis and Y-chromosome microdeletions and cystic fibrosis mutations tests) [22]. All patients underwent two consecutive semen analyses at least 3 months apart [8]; semen samples were collected by masturbation after a sexual abstinence of 2–7 days and analysed within 60 minutes of ejaculation, in accordance with the WHO criteria.

The improved Neubauer hemocytometer chamber (100-μm-deep; Brand™ Blaubrand™ Neubauer Improved Counting Chambers, Fisher Scientific, Loughborough, UK) was used for the calculation of sperm concentration and total sperm count in the ejaculate. Sperm morphology was assessed through the following steps: preparation of a smear of semen on a slide; fixing and staining the slide (Testsimplets® Prestained Slides, Waldeck GmbH & Co. KG, Münster, Germany); examination with brightfield optics at ×1000 magnification (Nikon Eclipse E 200, Nikon Instruments Europe B.V., Rome, Italy) with oil immersion; and assessment of approximately 200 spermatozoa per replicate for the percentage of normal or abnormal forms. Sperm motility was assessed by mixing twice the sample, using a wet preparation of 20 microM deep for each replicate, by examining the slide with phase-contrast optics at ×200 magnification and by assessing approximately 200 spermatozoa per replicate for the percentage of different motile categories.

In the laboratory for semen analysis a continuous quality assurance programme has been developed and maintained for several years. It relies on a quality manual containing standard operating procedures (SOP) and a detailed set of instructions for the different processes and

methods used in the laboratory. Internal quality control (IQC) is implemented with the inclusion of IQC materials in the laboratory's regular workload and the results for these materials are monitored using quality control charts. External quality control (EQA) is regularly performed through peer comparison and proficiency testing programmes (Italian EQA program). Results are sent to a central facility that assesses the performance of the laboratory. Continuous training and education of the laboratory personnel is also undertaken.

For the specific purposes of this study, we considered semen volume, sperm concentration, progressive sperm motility and morphology. Total motile sperm count was calculated and further categorized according to Hamilton et al [23]. According to WHO reference criteria, oligozoospermia was defined as <15 million sperm per mL; asthenozoospermia as <32% progressive motility, and teratozoospermia as <4% of normal forms. Likewise, oligo-astheno-teratozoospermia (OAT) was defined when all three anomalies occur simultaneously and azoospermia was considered as the complete absence of spermatozoa in semen after centrifugation [8]. The same laboratory was used for analyses of all parameters. We excluded 204 (13.1%) men because they missed one or more of the entry criteria [i.e., abnormal genetic tests (any type) ($n$ = 32; 2.0%); symptoms suggestive for genitourinary infections ($n$ = 21; 1.3%) or positive semen or urine cultures ($n$ = 146; 9.3%); a history of assisted reproductive techniques (ART) (any type) during the preceding year ($n$ = 5; 0.3%)]. A convenient sample of 1358 infertile men was considered for the final statistical analyses. Data collection followed the principles outlined in the Declaration of Helsinki. All men signed an informed consent agreeing to share their own anonymous information for future studies. The study was approved by our Hospital Ethical Committee (Prot. 2014—Pazienti Ambulatoriali).

## Statistical methods

Distribution of data was tested with the Shapiro–Wilk test. Data are presented as medians (interquartile range; IQR) or frequencies (proportions). The analyses consisted of several statistical steps. First, we considered only infertile men with a first semen sample with results above WHO references criteria ($n$ = 212, 15.6%). Second, in this cohort, the Mann–Whitney test and the Chi-square test were used to compare baseline clinical and demographics characteristics, hormonal values, and semen parameters between those individuals with a second semen analysis above (Group 1) vs. below (Group 2) WHO limits. Third, univariable and multivariable logistic regression analyses tested the association between clinical, hormonal, and seminal characteristics (i.e., age, CCI, FSH, TV and sperm concentration on the first sample) with second semen analysis below WHO limits after a first test above limits. Receiver Operating Characteristic (ROC) curves were generated to find TV, FSH and sperm concentration threshold values (defined as Youden J Index) to predict a second semen analysis below WHO limits. Fourth, these factors were used to create a new data-driven predictive model for sperm parameters below limits at a second test. Considering 1-point for each positive variable of the model, we calculated the positive predictive value (PPV) for second semen analysis below WHO limits as a function of the number of positive variables. Fifth, decision curve analysis (DCA) was used to assess the clinical benefit of using the newly identified model in respect to EAU [1] and AUA/ASRM guidelines [2]. Lastly, the diagnostic accuracy of the new model was compared with the AUA/ASRM guidelines, using a DeLong test. Statistical analyses were performed using SPSS v.26 (IBM Corp., Armonk, NY, USA) and R (2019), a language and environment for statistical computing (R Foundation for Statistical Computing, Vienna, Austria). All tests were two sided and statistical significance level was determined at p < 0.05.

## Results

Of 1358, 212 (15.6%) men had semen parameters above WHO limits at first assessment. Of those, 87 (41.0%) had a second semen analysis above WHO limits, while 80 (37.7%), 35 (16.5%) and 10 (4.7%) men showed 1, 2 and 3 sperm characteristics below limits at second test, respectively. Of note, the ejaculatory abstinence time was similar between the two semen analyses.

Table 1 details descriptive statistics of the entire cohort of men. Moreover, Table 1 also details baseline descriptive statistics of the sub cohort of men ($n$ = 212) with two consecutive semen analyses above limits (Group 1) compared to patients with a first above and a second below WHO limits (Group 2). Of this sub cohort, men in Group 2 had higher CCI and FSH values, but lower TV (all p≤0.04) as compared with than those in Group 1. Despite within normal ranges in both groups, men in Group 1 had greater total sperm number and TMSC even at first assessment compared to patients in Group 2 (p<0.001). For each group, the individual abstinence time between the first and second test was similar [mean (standard deviation) difference: Group1: 0.32 (0.2) days; Group 2: 0.33 (0.2) days]; similarly, abstinence time was not different among groups.

Table 2 reports logistic regression models predicting semen analysis below WHO limits after a first test above limits. At multivariable logistic regression analysis, lower TV (OR 0.9, p = 0.03), higher FSH values (OR 1.2, p<0.01) and lower total sperm number at first semen analysis (OR 0.9, p<0.01) were associated with second semen analyses below WHO limits, after accounting for age and CCI.

ROC curves showed that TV <15 ml, FSH values >6 mUI/ml and total sperm number $< 120$ x$10^6$ had good predictive ability for sperm parameters below WHO limits at second analysis (all AUC >0.7). Thereafter, a risk score for second semen sample results below limits using the three previous variables was developed. Considering 1-point for each of the previous positive variable of the model, PPV of a second semen analysis below WHO limits increased from 38.5% to 74.4%, 77.1% and 100% among patients with risk scores of 0, 1, 2 and 3, respectively (p<0.001). The DeLong's test confirmed that the new model (AUC 0.73, 95% CI: 0.56–0.71) performed better than the AUA guidelines (AUC 0.64, 95%CI: 0.69–0.77) (p<0.001).

Lastly, DCA (Fig 1) displays the superior net benefit of using the new model in terms of predicting a second semen analysis below WHI limits, compared to the EAU and AUA/ASRM guidelines recommendations.

## Discussion

The lack of consensus among international guidelines on the issue prompted us to analyse findings from a relatively large homogeneous cohort of primary infertile men with the specific aim of identifying patients at higher risk of having a semen analysis below WHO limits following a first investigation above limits. Of clinical importance, we found that approximately 60% of infertile men with a first semen analysis above limits depicted a second test below WHO limits [8]. Patients with lower TV, higher FSH values and lower total sperm number at the first semen analysis emerged to be at greater risk of subsequent semen samples below WHO limits, thus deserving a confirmatory analysis. A new risk score based on these associated factors was then compared to the AUA/ASRM guidelines recommendation to test all infertile men with two consecutive semen samples. We showed the net benefit of the new risk score in respect to the AUA/ASRM guidelines in identifying infertile men with a second semen sample with results below limits after a first one above limits. Therefore, the proposed risk score could be used in clinical practice for the management of infertile men with a first semen analysis above

**Table 1. Descriptive statistics of the whole cohort of infertile men and according to second semen analysis characteristics.**

| | Overall | Group 1 | Group 2 | p-value[*] |
|---|---|---|---|---|
| No. of individuals | 1358 | 87 (41.0%) | 125 (59.0%) | |
| Age (years) | | | | 0.6 |
| Median (IQR) | 37.0 (34–40) | 37.0 (34–41) | 37.0 (34–41) | |
| Range | 18–55 | 20–54 | 25–54 | |
| Duration of infertility (months) | | | | 0.4 |
| Median (IQR) | 24 (12–36) | 24.0 (12–36) | 24.0 (15–36) | |
| Range | 12–228 | 12–135 | 24–108 | |
| BMI (kg/m$^2$) | | | | 0.9 |
| Median (IQR) | 24.9 (23.2–26.8) | 24.6 (23.4–25.9) | 24.4 (23.0–26.7) | |
| Range | 18.5–41.2 | 19.8–39.2 | 18.6–36.8 | |
| CCI (score) | | | | 0.04 |
| Median (IQR) | 0.0 (0.0) | 0.0 (0.0) | 0.0 (0.0) | |
| Mean (SD) | 0.1 (0.2) | 0.01 (0.1) | 0.2 (0.1) | |
| Range | 0–8 | 0–1 | 0–4 | |
| CCI ≥ 1 [No. (%)] | 84 (6.2) | 1 (1.1) | 9 (7.2) | 0.04 |
| Current smoking status [No. (%)] | 387 (28.5) | 20 (23.0) | 33 (26.4) | 0.6 |
| Mean TV (Prader's estimation) | | | | <0.001 |
| Median (IQR) | 18.0 (14–20) | 23.0 (18–25) | 18.0 (13–23) | |
| Range | 4–25 | 8–25 | 8–25 | |
| Varicocele [No. (%)] | 821 (60.5) | 49 (56.3) | 69 (55.2) | 0.8 |
| tT (ng/mL) | | | | 0.9 |
| Median (IQR) | 4.7 (3.6–5.9) | 4.8 (3.7–6.4) | 4.8 (3.7–5.9) | |
| Range | 0.9–23.5 | 1.8–9.9 | 2.0–13.4 | |
| FSH (mUI/mL) | | | | <0.001 |
| Median (IQR) | 4.6 (3.0–7.6) | 3.3 (2.1–4.6) | 4.7 (2.9–6.8) | |
| Range | 0.8–45.8 | 0.8–15.3 | 1.2–15.0 | |
| LH (mUI/mL) | | | | 0.3 |
| Median (IQR) | 3.8 (2.7–5.2) | 3.0 (2.3–4.5) | 3.4 (2.5–4.4) | |
| Range | 0.9–34.2 | 0.8–8.7 | 1.1–12.4 | |
| Prolactin (ng/mL) | | | | 0.9 |
| Median (IQR) | 8.2 (6.2–11.5) | 8.1 (6.0–12.4) | 8.2 (6.3–11.1) | |
| Range | 1.9–34.3 | 3.4–22.4 | 2.6–32.3 | |
| SHBG (nmol/L) | | | | 0.1 |
| Median (IQR) | 32.0 (24–41) | 35.8 (24–43) | 29.0 (22–40) | |
| Range | 9.0–85.0 | 13.0–57.0 | 9.0–85.0 | |
| TSH (mUI/L) | | | | 0.5 |
| Median (IQR) | 1.6 (1.1–2.3) | 1.6 (1.1–2.7) | 1.6 (1.2–2.1) | |
| Range | 0.5–6.6 | 0.5–4.2 | 0.6–5.4 | |
| Sexual abstinence (days) | | | | 0.7 |
| Median (IQR) | 3.0 (2.0–5.0) | 3.0 (2.0–5.0) | 3.0 (2.0–5.0) | |
| Range | 2.0–7.0 | 2.0–7.0 | 2.0–7.0 | |
| Semen volume (mL) | | | | 0.8 |
| Median (IQR) | 3.0 (2.0–4.0) | 3.0 (2.0–4.0) | 3.0 (2.0–4.0) | |
| Range | 1.0–11–0 | 1.0–8.0 | 1.0–8.0 | |
| Sperm concentration (x10$^6$/mL) | | | | <0.001 |
| Median (IQR) | 18.1 (5.0–42.1) | 62.0 (35–94) | 34.0 (24–58) | |
| Range | 0.1–455.3 | 16.0–455.3 | 15.0–198.0 | |

*(Continued)*

**Table 1.** (Continued)

| | Overall | Group 1 | Group 2 | p-value* |
|---|---|---|---|---|
| Total sperm number (x10$^6$ per ejaculate) | | | | <0.001 |
| Median (IQR) | 52.0 (14–142) | 173.6 (118–282) | 120.0 (80–180) | |
| Range | 0.1–992.0 | 16.5–896.0 | 18.0–992.0 | |
| Progressive sperm motility (%) | | | | 0.2 |
| Median (IQR) | 25.0 (10–39) | 46.5 (40–56) | 44.0 (38–53) | |
| Range | 0.0–96.0 | 32.0–76.0 | 32.0–85.0 | |
| Normal sperm morphology (%) | | | | 0.9 |
| Median (IQR) | 3.0 (1–12) | 13.0 (6–36) | 12.0 (8–30) | |
| Range | 0.0–100.0 | 4.0–94.0 | 4–90.0 | |
| TMSC (x10$^6$) | | | | <0.001 |
| Median (IQR) | 10.4 (1.2–40.1) | 82.2 (52.4–126.8) | 52.8 (30.5–82.7) | |
| Range | 0.0–615.1 | 0.0–492.8 | 7.2–615.1 | |
| TMSC groups [No. (%)] | | | | 0.2 |
| < 5 x106 | 533 (39.2) | – | – | |
| 5–20 x106 | 287 (21.1) | 2 (2.3) | 7 (5.6) | |
| > 20 x106 | 538 (39.6) | 85 (97.7) | 118 (94.4) | |
| SDF (%) | | | | 0.8 |
| Median (IQR) | 32.8 (19.5–48.6) | 26.4 (10–39) | 27.5 (14–54) | |
| Range | 0.4–97.7 | 7.0–72.0 | 5.0–93.0 | |

Keys: Group 1: infertile men with two consecutive semen samples with results above WHO limits; Group 2: infertile men with a first semen analysis above WHO limits and a second test below WHO limits.

BMI = body mass index; CCI = Charlson Comorbidity Index; TV = testicular volume; tT = total Testosterone; FSH = follicle-stimulating hormone, LH = luteinizing hormone SHBG = Sex hormone binding globulin; TSH = Thyroid-stimulating hormone; TMSC = Total Motile Sperm Count SDF = Sperm DNA fragmentation Index.

* p value according to the Mann-Whitney test and Chi Square test, as indicated.

WHO limits. It is well known that semen analysis per se is a macroscopic evaluation of sperm characteristics and does not necessarily and reliably distinguish between fertile and infertile men [5]. Moreover, semen parameters are subject to a certain degree of intra-individual variability over time [14]. Consequently, at least two consecutive semen analyses are usually requested for the initial management of infertile couples in clinical practice. While this approach is certainly endorsed in case of sperm parameters with results below WHO limits at baseline investigation, this is not the case for infertile men presenting with a first semen

**Table 2. Logistic regression models predicting semen analysis below WHO limits after a first test above limits (No. = 212).**

| | UVA model | | | MVA model | | |
|---|---|---|---|---|---|---|
| | OR | p-value | 95% CI | OR | p-value | 95% CI |
| Age | 0.97 | 0.4 | 0.92–1.03 | 0.97 | 0.5 | 0.90–1.07 |
| CCI | 4.25 | 0.1 | 0.75–15.82 | 2.82 | 0.2 | 0.51–12.29 |
| TV | 0.88 | <0.01 | 0.84–0.94 | 0.93 | 0.03 | 0.86–0.99 |
| FSH | 1.31 | <0.01 | 1.13–1.51 | 1.22 | <0.01 | 1.10–1.47 |
| Total sperm number* | 0.97 | <0.001 | 0.95–0.99 | 0.97 | <0.01 | 0.94–0.99 |

Keys: UVA = Univariate model; MVA = Multivariate model, CCI = Charlson Comorbidity Index

TV = testicular volume; FSH = follicle-stimulating hormone, LH = luteinizing hormone

*at first semen analysis

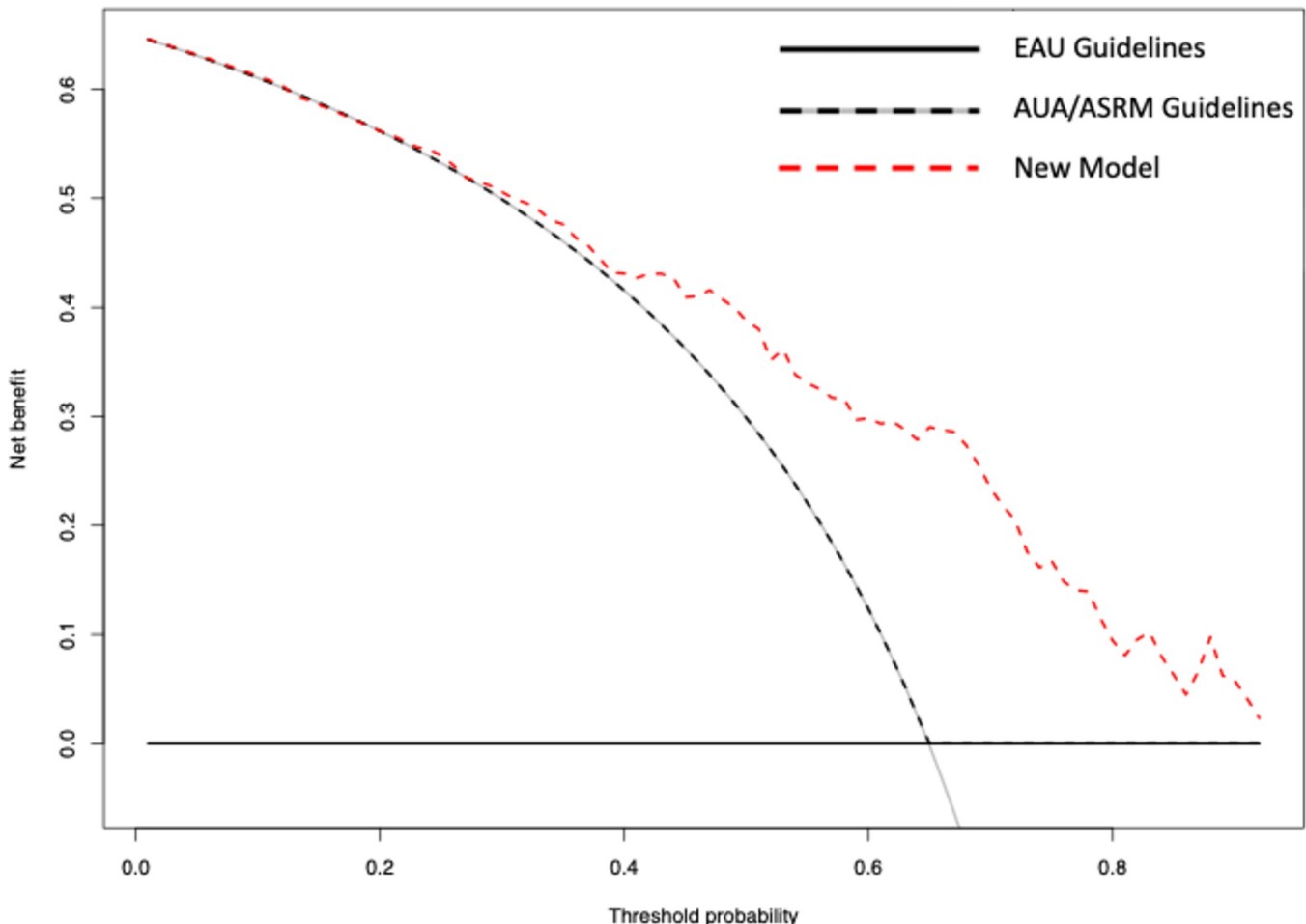

**Fig 1. Decision curve analysis.** Decision curve analysis showing the net benefit of using the new data-driven model to identify patients with a first normal semen sample who should be tested for a potential pathological second semen analysis. The black solid line represents the strategy of screening none of the patients (EAU model); the black dashed and grey solid line represents the strategy of screening all patients (AUA/ASRM model); the red dashed line represents the strategy of screening patients according to the new model.

analysis above limits. Indeed, current EAU and AUA/ASRM Guidelines report conflicting recommendations regarding the need for a confirmatory semen analysis in infertile men with a first test above WHO limits [1, 2]. This criticism is of primary clinical importance because unnecessary additional samples from men with a first semen analysis above WHO limits could lead to increasing costs and frustration for the couple; on the contrary, missing a second semen samples with results below limits could delay further diagnostics tests and a tailored treatment. Likewise, exclusively relying on a single semen analysis during the diagnostic work-up of infertile men may also increase the risk of referring couples to ART even without receiving an adequate baseline andrological evaluation [24]. That said, it is becoming worldwide endorsed the idea that reference ranges and reference limits for semen parameters are useful for research purposes, to guide diagnostic testing and treatment choice, but decision limits based on clinical and statistical considerations should be used in clinical practice for the management of infertile men [25].

Previous studies have investigated the rate of concordance between consecutive semen analyses and the need for a second test in healthy sperm donors and infertile men. A

population based-study with 998 volunteers aged 20–60 years, analysed data from 332 men who provided single semen samples and 666 men who provided two samples [11]. Authors found that semen parameters were similar between the one-sample group and the two-sample group; moreover, the difference in semen parameters between first and second samples were relatively small. Despite some intra-individual variations of sperm quality, one ejaculate was deemed as a sufficient indicator of semen quality in epidemiological studies [11]. Leushuis et al. [12], for instance, described the within-individual variability of semen analyses in male partners of subfertile couples in a large retrospective cohort. They showed a high within-subject variability for sperm motility and morphology, thus questioning the classification of man based on a single semen measurement. The Environment and Reproductive Health (EARTH) Study investigated the long-term variability in semen parameters in a cohort of 329 infertile men who provided 768 semen samples [13]. Authors showed no significant differences between mean values of first samples and the means of the remaining samples. However, when semen parameters were dichotomized, a second sample was necessary to estimate the prevalence of pathological results according to WHO reference criteria [13]. Of note, 51% of men with sperm parameters above WHO limits at first assessment had at least one semen parameter below the WHO reference limits in their second sample [13]. More recently, Blickenstorfer et al. [10] analysed data from 2,566 infertile men who underwent at least two consecutive semen analyses and showed that 51% of the second analyses confirmed the initial findings; conversely, 27% of men who presented with normozoospermia in the first semen sample had a second sample with results below limits. Finally, they reported a limited discriminating capacity of each semen parameter to distinguish between men with a second semen analysis above vs. below WHO limits. Overall, our results confirm the latter mentioned findings, since we showed that 60% of infertile men with a first semen analysis above WHO limits had a second test below limits. In addition, we found that lower TV, lower total sperm number and higher FSH values were significantly associated with the risk of having a semen test below limits. The clinical implication of our study is several-fold. First, this is the first study to establish a novel risk model for the selection of infertile men presenting with semen analysis above WHO limits at first assessment that would deserve a confirmatory test because at higher risk of a second semen analysis below limits. Of clinical importance, the novel score is user-friendly since it is based on clinical, hormonal and sperm parameters that are essential components of the baseline diagnostic work up of every infertile man. As compared with previous data, current findings are even more clinically relevant since we had investigated a relatively large cohort of same ethnicity, non-Finnish white-European primary infertile men with an identical thorough clinical, hormonal and semen evaluation. Conversely, most of the previous studies had investigated the association and predicted ability of consecutive semen parameters while ignoring clinical and laboratory features of their cohorts [10, 13], thus potentially limiting the clinical validity of their findings in the real-life setting. Likewise, we showed the net benefit of applying a personalised approach to identify those men with a second semen test below WHO limits, based on the specific newly developed risk score as compared with the AUA/ASRM Guidelines model of testing all. The implementation of our risk score in clinical practice might spare unnecessary test and avoid delaying in treatment for a specific cohort of infertile men with semen analysis above WHO limits. As a whole, our results strongly outline the compulsory need that every infertile man gets a comprehensive evaluation by specialists in male reproduction [1, 2, 24], including those with a first semen analysis above WHO limits. In fact, on the one hand, approximately 60% of them will show semen parameters below WHO limits at a second test, and on the other, semen analysis per se cannot be relied on to rule out a male factor cause of infertility. Our study is not devoid of limitations. First, even though we analysed a relatively large, homogeneous, same-ethnicity cohort of infertile men, this was a single centre-

based study, raising the possibility of selection biases; thereof, larger studies across different centres and cohorts are needed to externally validate our findings. Second, despite all semen analyses have been performed at the same laboratory according to WHO reference criteria [8], it is not possible to completely exclude a partial inter-observers variability, which surely cannot be considered sufficient to explain such an impressive 60% of discrepancy between consecutive semen samples in the same patient.

## Conclusions

In this cross-sectional study we found that approximately 60% of infertile men with a first semen analysis above WHO limits depicted a second test with results below limits. Lower testicular volume, higher FSH values and lower total sperm number at first analysis were independently associated with a second semen analysis below limits. Thereof, we proposed a novel and highly performing risk score to identify infertile men at greater risk of having a second semen sample with results below WHO limits after a first one above limits. This score could be used in clinical practice to better tailor the management of men presenting for couple's infertility despite having initial semen samples above WHO limits, while avoiding unnecessary examination and delays in treatment.

## Supporting information

**S1 Dataset.**
(XLSX)

## Author Contributions

**Conceptualization:** Luca Boeri, Edoardo Pozzi.

**Data curation:** Luca Boeri, Edoardo Pozzi, Paolo Capogrosso, Giuseppe Fallara, Federico Belladelli, Luigi Candela, Nicolò Schifano, Christian Corsini, Walter Cazzaniga, Daniele Cignoli, Eugenio Ventimiglia, Marina Pontillo, Massimo Alfano.

**Formal analysis:** Luca Boeri, Edoardo Pozzi.

**Investigation:** Luca Boeri, Edoardo Pozzi, Andrea Salonia.

**Methodology:** Edoardo Pozzi, Francesco Montorsi, Andrea Salonia.

**Project administration:** Luca Boeri, Francesco Montorsi, Andrea Salonia.

**Supervision:** Francesco Montorsi, Andrea Salonia.

**Validation:** Francesco Montorsi, Andrea Salonia.

**Visualization:** Andrea Salonia.

**Writing – original draft:** Luca Boeri, Edoardo Pozzi.

**Writing – review & editing:** Andrea Salonia.

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
