## [Decision Letter · Decision Letter 0]

17 Jun 2022

PONE-D-22-12621Infertile men with normal semen parameters at first assessment may deserve a second semen analysis: challenging the guidelines in the real-life scenarioPLOS ONE

Dear Dr. Salonia,

Thank you for submitting your manuscript to PLOS ONE. After careful consideration, we feel that it has merit but does not fully meet PLOS ONE’s publication criteria as it currently stands. Therefore, we invite you to submit a revised version of the manuscript that addresses the points raised during the review process.

We look forward to receiving your revised manuscript.

Kind regards,

Stefan Schlatt

Academic Editor

PLOS ONE

Journal Requirements:

3. Please include your tables as part of your main manuscript and remove the individual files. Please note that supplementary tables (should remain/ be uploaded) as separate "supporting information" files

Additional Editor Comments:

This study aims to describe differences between a primary and secondary semen analysis for evlautaion of infertility. The authors assess a second sample from subjects who had shown normal values in accordance to current guidelines. Depending on the likelihood of an abnormal second semen

analyses the authors propose a new predictive model to better identify subjects with high risk of pathological changes despite a normal first ejaculate score. The authors describe limitations of their work and the paper can add to discussions on improvement of guidelines and recommendations. The referee made several important comments which affect the validity of the findings. The authors need to address these comments and should revise thier manuscript accordingly.

Reviewers' comments:

Reviewer's Responses to Questions

**Comments to the Author**

1. Is the manuscript technically sound, and do the data support the conclusions?

Reviewer #1: Partly

2. Has the statistical analysis been performed appropriately and rigorously? 

Reviewer #1: Yes

3. Have the authors made all data underlying the findings in their manuscript fully available?

Reviewer #1: No

4. Is the manuscript presented in an intelligible fashion and written in standard English?

Reviewer #1: Yes

5. Review Comments to the Author

Reviewer #1: There is no comprehensive description of how the basic semen analysis was performed. Data on variability and compliance with e.g. demand for comparison of replicate assessments, use of equipment able to provide realiable results, interntal and external quality control is essential for this study.

Furthermore, although it is true that there is definition of infertility based on TTP=< 12 months, but this has been challenged not the least based on older comprehensive studies (Tietze 1968 e.g.) and recent studied showing huge differences between short time to pregnancy and WHO "reference limits" - and those "reference limits" have also been challenged in the most recent WHO manual (2022).

Authors claim semen assessments were done "within two hours". This is clearly NOT in compliance with WHO 2010: page 21, 2.5 Sperm Motility second paragraph: "Sperm motility within semen should be assessed as soon as possible after liquefaction of the sample, preferably at 30 minutes, but in any case within 1 hour, following ejaculation, to limit the deleterious effects of dehydration, pH or changes in temperature on motily."

This calls in question to what extent reliable laboratory methods have been used. Lack of compliance with WHO recommendations can indicate that the assessments are influenced by significant technical variability - and that the reported change from "normal" to "abnormal" to a significant degree depend on technical /laboratory/ errors.

6. PLOS authors have the option to publish the peer review history of their article (what does this mean?). If published, this will include your full peer review and any attached files.

Reviewer #1: No

---

## [Author Response · Author response to Decision Letter 0]

29 Jul 2022

Dr. Emily Chenette, PhD

Editor-in-Chief, PLOS ONE

Dr. Stefan Schlatt

Milan, July 14th, 2022

Dear Dr Chenette,

dear Dr. Schlatt,

please find enclosed the revised version of the manuscript titled “Infertile men with normal semen parameters at first assessment may deserve a second semen analysis: challenging the guidelines in the real-life scenario” (PONE-D-22-12621- Authors: Luca Boeri, et al.) to be considered for publication in PLOS ONE.

We are very grateful to the Reviewers for their insightful comments to our paper.

List of the changes made in the manuscript:

REVIEWER #1

COMMENT#1. 

There is no comprehensive description of how the basic semen analysis was performed. Data on variability and compliance with e.g. demand for comparison of replicate assessments, use of equipment able to provide reliable results, internal and external quality control is essential for this study.

A1. We thank the Reviewer#1 for this insightful comment. We have included a detailed description of basic semen analysis and quality control methods in the Methods section of the manuscript.

COMMENT#2. 

Furthermore, although it is true that there is definition of infertility based on TTP=< 12 months, but this has been challenged not the least based on older comprehensive studies (Tietze 1968 e.g.) and recent studied showing huge differences between short time to pregnancy and WHO "reference limits" - and those "reference limits" have also been challenged in the most recent WHO manual (2022).

A2. We thank the Reviewer#1 for this comment. It is well known that infertility is not related to the mere sperm quality, but we relied on the WHO definition of infertility, which is the one used by most of the international scientific societies (e.g., European Association of Urology Guidelines; American Society for Reproductive Medicine).

COMMENT#3. 

Authors claim semen assessments were done "within two hours". This is clearly NOT in compliance with WHO 2010: page 21, 2.5 Sperm Motility second paragraph: "Sperm motility within semen should be assessed as soon as possible after liquefaction of the sample, preferably at 30 minutes, but in any case within 1 hour, following ejaculation, to limit the deleterious effects of dehydration, pH or changes in temperature on motily."

A3. We thank the Reviewer#1 for this critical comment. During the revision of our manuscript, we have double checked once more the whole procedure with our laboratory biologists and indeed we may confirm that all samples have been and are rigorously analysed within 60 min. from collection. The text has been revised accordingly.

COMMENT#4. 

This calls in question to what extent reliable laboratory methods have been used. Lack of compliance with WHO recommendations can indicate that the assessments are influenced by significant technical variability - and that the reported change from "normal" to "abnormal" to a significant degree depend on technical /laboratory/ errors.

A4. As comprehensively detailed before, our laboratory analyzed all semen samples strictly according to WHO indications. We have also included internal and external quality control throughout the Methods section of the manuscript. 

We hope that the paper is now suitable to be considered for publication in the Original Articles section of PLOS ONE.

Sincerely yours,

Andrea Salonia, on behalf of all the authors

Andrea Salonia, MD, PhD, FECSM

University Vita-Salute San Raffaele

Division of Experimental Oncology/Unit of Urology, URI-Urological Research Institute

IRCCS Ospedale San Raffaele

Email: salonia.andrea@hsr.it

---

## [Decision Letter · Decision Letter 1]

30 Aug 2022

PONE-D-22-12621R1Infertile men with normal semen parameters at first assessment may deserve a second semen analysis: challenging the guidelines in the real-life scenarioPLOS ONE

Dear Dr. Salonia,

Thank you for submitting your manuscript to PLOS ONE. After careful consideration, we feel that it has merit but does not fully meet PLOS ONE’s publication criteria as it currently stands. Therefore, we invite you to submit a revised version of the manuscript that addresses the points raised during the review process.

We look forward to receiving your revised manuscript.

Kind regards,

Stefan Schlatt

Academic Editor

PLOS ONE

Journal Requirements:

Additional Editor Comments:

The reviewer requests raises a few concerns primarily in regard to the interpretation of the data. The authors may address these concerns in a revised version of the manuscript.

Reviewers' comments:

Reviewer's Responses to Questions

**Comments to the Author**

1. If the authors have adequately addressed your comments raised in a previous round of review and you feel that this manuscript is now acceptable for publication, you may indicate that here to bypass the “Comments to the Author” section, enter your conflict of interest statement in the “Confidential to Editor” section, and submit your "Accept" recommendation.

Reviewer #1: (No Response)

2. Is the manuscript technically sound, and do the data support the conclusions?

Reviewer #1: Partly

3. Has the statistical analysis been performed appropriately and rigorously? 

Reviewer #1: I Don't Know

4. Have the authors made all data underlying the findings in their manuscript fully available?

Reviewer #1: No

5. Is the manuscript presented in an intelligible fashion and written in standard English?

Reviewer #1: Yes

6. Review Comments to the Author

Reviewer #1: I thank the authors for clarifying a number of matters.

Still I think there are unclear items to work with.

1. Authors only work with sperm concentration, but declares sperm numbers were calculated. It was clear already 2010 that the WHO manual recommends sperm number rather than concentration that depends on both sperm number and dilution by fluid from other sources.

2. Variation in ejaculatory abstinence time can be an important source for variation. There is not even a discussion about this and no data available, only in M&M that abstinence time was 2-7 days. Could there be individual variability due to variation in abstinence time (or even ejaculatory frequency) before the two semen analyses?

3. Have the authors considered that limits above the lower 5th percentile "limit" byt the WHO 2010 and 2021 could be better to distinguish between men with semen abnormalities and men without? Looking at e.g. Keihani et al 2021 indicating that men in couples with short time to initiation of pregnancy have much higher semen analysis results compared to men in couples with 5-12 months.

Thus, I do agree with the authors that the so called WHO limits (which should not be mistaken for limits between fertile and subfertile men) do mean that there is a risk to miss men with fertility problems. What I miss (or have missed) is a discussion whether higher DECISION LIMITS could be an alternative solution to repeating semen analysis

7. PLOS authors have the option to publish the peer review history of their article (what does this mean?). If published, this will include your full peer review and any attached files.

Reviewer #1: No

---

## [Author Response · Author response to Decision Letter 1]

18 Sep 2022

Dr. Emily Chenette, PhD

Editor-in-Chief, PLOS ONE

Dr. Stefan Schlatt

Milan, September 18th, 2022

Dear Dr Chenette,

dear Dr. Schlatt,

please find enclosed the revised version of the manuscript titled “Infertile men with normal semen parameters at first assessment may deserve a second semen analysis: challenging the guidelines in the real-life scenario” (PONE-D-22-12621R1- Authors: Luca Boeri, et al.) to be considered for publication in PLOS ONE.

We are very grateful to the Reviewers for their insightful comments to our paper.

List of the changes made in the manuscript:

Journal Requirements

A1. The whole Reference list has been reviewed carefully; no papers have been retracted. 

REVIEWER #1

COMMENT#1. 

Authors only work with sperm concentration, but declares sperm numbers were calculated. It was clear already 2010 that the WHO manual recommends sperm number rather than concentration that depends on both sperm number and dilution by fluid from other sources.

A1. We thank the Reviewer#1 for this critical comment. Accordingly, total sperm number data has been included in the Results section of the manuscript. Thus, we have included total sperm number instead of sperm concentration in the analyses and previous findings were confirmed. The text has been revised accordingly. 

COMMENT#2. 

Variation in ejaculatory abstinence time can be an important source for variation. There is not even a discussion about this and no data available, only in M&M that abstinence time was 2-7 days. Could there be individual variability due to variation in abstinence time (or even ejaculatory frequency) before the two semen analyses? 

A2. Ejaculatory abstinence time was similar between the two semen analyses [Median (IQR) 3 (1-4) days]. The Results section of the manuscript has been revised, accordingly.

COMMENT#3. 

Have the authors considered that limits above the lower 5th percentile "limit" byt the WHO 2010 and 2021 could be better to distinguish between men with semen abnormalities and men without? Looking at e.g. Keihani et al 2021 indicating that men in couples with short time to initiation of pregnancy have much higher semen analysis results compared to men in couples with 5-12 months. Thus, I do agree with the authors that the so called WHO limits (which should not be mistaken for limits between fertile and subfertile men) do mean that there is a risk to miss men with fertility problems. What I miss (or have missed) is a discussion whether higher DECISION LIMITS could be an alternative solution to repeating semen analysis

A3. We thank the Reviewer#1 for this insightful comment. We agree with the Reviewer#1 and the WHO2021 suggesting that semen parameters per se cannot distinguish between fertile vs. infertile men. However, here we included infertile men only as for WHO definition (i.e., not conceiving after at least 12 months of unprotected intercourses regardless of whether or not a pregnancy ultimately occurs) which is actually independent from semen analysis parameters. Thus, semen parameters are used in clinical practice to guide diagnosis testing and the choice of treatment options for infertile men. Therefore, we aimed to challenge international guidelines concerning the need for a single or confirmative semen analysis. The need to use decision limits in clinical practice has been included in the Discussion section of the manuscript. 

We hope that the paper is now suitable to be considered for publication in the Original Articles section of PLOS ONE.

Sincerely yours,

Andrea Salonia, on behalf of all the authors

Andrea Salonia, MD, PhD, FECSM

Vita-Salute San Raffaele University

Division of Experimental Oncology/Unit of Urology, URI-Urological Research Institute

IRCCS Ospedale San Raffaele

Email: salonia.andrea@hsr.it

---

## [Decision Letter · Decision Letter 2]

2 Nov 2022

PONE-D-22-12621R2Infertile men with normal semen parameters at first assessment may deserve a second semen analysis: challenging the guidelines in the real-life scenarioPLOS ONE

Dear Dr. Salonia,

Thank you for submitting your manuscript to PLOS ONE. After careful consideration, we feel that it has merit but does not fully meet PLOS ONE’s publication criteria as it currently stands. Therefore, we invite you to submit a revised version of the manuscript that addresses the points raised during the review process.

We look forward to receiving your revised manuscript.

Kind regards,

Stefan Schlatt

Academic Editor

PLOS ONE

Journal Requirements:

Additional Editor Comments:

The reviewer and I are positive about the content of the paper and considered the revisions useful. However, the refree still raises a few minor concerns which in my view are relevant and indeed improve the papers validity. The data need to be presented in a format that can easily be readable and contains all labels. I ask the authors to once more go through the suggestions and fix the errors listed by the reviewer.

Reviewers' comments:

Reviewer's Responses to Questions

**Comments to the Author**

1. If the authors have adequately addressed your comments raised in a previous round of review and you feel that this manuscript is now acceptable for publication, you may indicate that here to bypass the “Comments to the Author” section, enter your conflict of interest statement in the “Confidential to Editor” section, and submit your "Accept" recommendation.

Reviewer #1: (No Response)

2. Is the manuscript technically sound, and do the data support the conclusions?

Reviewer #1: Partly

3. Has the statistical analysis been performed appropriately and rigorously? 

Reviewer #1: No

4. Have the authors made all data underlying the findings in their manuscript fully available?

Reviewer #1: No

5. Is the manuscript presented in an intelligible fashion and written in standard English?

Reviewer #1: No

6. Review Comments to the Author

Reviewer #1: I do believe this study merits to be published but there are still a number of aspects that if considered could make this a landmark publication. There are also flaws that need to be corrected.

1. The supporting information is not sufficiently transparent.

a. Many labels and contents are in Italian - must be in English

b. Many cells have only an Error indication

c. A complete lack of explanations of column labels, how the reader should be able to analyse the data (e.g. compare first and second sample

2. The systematic use of terms like "normal semen" and "abnormal semen" although even WHO has pointed out that data should not be interpreted in that way: reference limits in WHO 2010 and 2021 are NOT limits between fertility and infertility. It would be better to refer to "men with semen results below/above WHO reference limts"

WHO 2021 even points to the fact that male factor infertility have different causes, and therefore it is not very likely that there exists ONE true limit concerning semen analysis results for all causes - decision limits are more likely to support the clinical work with male reproductive disorders.

3. In table one, authors show the range of a number semen analysis parameters. It is strange that among men with "normal" first and second sample, low sperm numbers, concentrations, volumes etc are found!

4. Concerning the abstinence time - there are only results on group level available. What would be of interest is the comparison of each individuals abstinence time for the two samples (average difference and variability).

5. A very interesting aspect that has not been approached (as I can see it) is that the WHO reference limits 2010 and 2021 are too low! Againg thinking about e.g. Keihani 2021 this is not unlikely at all. The data the authors have could carry important information about this: could it be that using higher cut-offs would make a majority of men with first "normal" result to be above the WHO 2010/2021 level but under a tentative, higher cut-off? A graph showing the results of "normal-abnormal" men's first results in relation to cut-off level and compared to "normal-normal" men's

Of course, this would mean that the heavily promoted need to for a second semen analysis based on the often misunderstood WHO 2010/2021 reference limits becomes of less interest. But the matter deserves serious scientific attention not the least since the WHO 2021 points to the weaknesses of the presented reference limits.

6. Authors repeatevly use the term"infertile men", but the definition is not sufficiently described. How where female factors ruled out?

7. PLOS authors have the option to publish the peer review history of their article (what does this mean?). If published, this will include your full peer review and any attached files.

Reviewer #1: No

---

## [Author Response · Author response to Decision Letter 2]

13 Nov 2022

Dr. Emily Chenette, PhD

Editor-in-Chief, PLOS ONE

Dr. Stefan Schlatt

Milan, November 06th, 2022

Dear Dr Chenette,

dear Dr. Schlatt,

please find enclosed the revised version of the manuscript titled “Infertile men with semen parameters above WHO reference limits at first assessment may deserve a second semen analysis: challenging the guidelines in the real-life scenario” (PONE-D-22-12621R2- Authors: Luca Boeri, et al.) to be considered for publication in PLOS ONE.

We are very grateful to the Reviewers for their insightful comments to our paper.

List of the changes made in the manuscript:

REVIEWER #1

COMMENT#1. 

The supporting information is not sufficiently transparent.

a. Many labels and contents are in Italian - must be in English

b. Many cells have only an Error indication

c. A complete lack of explanations of column labels, how the reader should be able to analyse the data (e.g. compare first and second sample).

A1. We thank the Reviewer#1 for this comment. The supporting information file has been revised. 

COMMENT#2. 

The systematic use of terms like "normal semen" and "abnormal semen" although even WHO has pointed out that data should not be interpreted in that way: reference limits in WHO 2010 and 2021 are NOT limits between fertility and infertility. It would be better to refer to "men with semen results below/above WHO reference limts"

WHO 2021 even points to the fact that male factor infertility have different causes, and therefore it is not very likely that there exists ONE true limit concerning semen analysis results for all causes - decision limits are more likely to support the clinical work with male reproductive disorders.

A2. We thank the Reviewer#1 for this comment. The entire text has been revised accordingly. 

COMMENT#3. 

In table one, authors show the range of a number semen analysis parameters. It is strange that among men with "normal" first and second sample, low sperm numbers, concentrations, volumes etc are found!

A3. Table 1 reports baseline descriptive statistics of men according to their second semen analysis. In details Group 1 includes men with two consecutive semen analyses above WHO limits and Group 2 includes infertile men with a first semen sample above and a consecutive second one below limits. Of note, sperm parameters included in table 1 are baseline values (meaning at first assessment). The primary aim of table 1 is showing baseline characteristics of men to identify potential discrepancies within the two groups. Moreover, sperm concentration and volume were note lower in Group 1 compared to group 2. 

COMMENT#4. 

Concerning the abstinence time - there are only results on group level available. What would be of interest is the comparison of each individuals abstinence time for the two samples (average difference and variability).

A4. We thank the Reviewer#1 for this comment. We performed a Wilcoxon Signed Rank Test to assess potential difference in each individual abstinence time for the two samples. No statistical difference was noted. Difference in abstinence time for Group 1: average 0.32 days; variability 0.2; difference in abstinence time for Group 2: average 0.33 days; variability 0.2. The text has been revised accordingly. 

COMMENT#5. 

A very interesting aspect that has not been approached (as I can see it) is that the WHO reference limits 2010 and 2021 are too low! Againg thinking about e.g. Keihani 2021 this is not unlikely at all. The data the authors have could carry important information about this: could it be that using higher cut-offs would make a majority of men with first "normal" result to be above the WHO 2010/2021 level but under a tentative, higher cut-off? A graph showing the results of "normal-abnormal" men's first results in relation to cut-off level and compared to "normal-normal" men's

Of course, this would mean that the heavily promoted need to for a second semen analysis based on the often misunderstood WHO 2010/2021 reference limits becomes of less interest. But the matter deserves serious scientific attention not the least since the WHO 2021 points to the weaknesses of the presented reference limits.

A5. We thank the Reviewer#1 for this comment. We totally agree that reference limits should be taken with caution in clinical practice. Particularly due to the fact that they derived from the 5th percentile of distribution from a cohort of fertile men. However, the WHO reference criteria are commonly used in clinical practice and the proposal of new cut-offs are beyond the aim of our study. In this study we aimed to identify baseline characteristics of men with semen parameters above the WHO limits (currently used worldwide) to dictate a second test. We believe that our study has a strong implication and relevance in the everyday clinical practice. We will conduct a new study in the near future thanks to your suggestion to depict different scenarios with higher limits.

COMMENT#6. 

Authors repeatevly use the term"infertile men", but the definition is not sufficiently described. How where female factors ruled out?

A6. As reported in the Methods section, couple’s infertility was defined by WHO definition (not conceiving a pregnancy after at least 12 months of unprotected intercourses regardless of whether or not a pregnancy ultimately occurs). Moreover, each men underwent a comprehensive clinical, physical, hormonal, genetic and seminal investigation to identify the cause of male infertility. Male factor infertility was defined after a comprehensive diagnostic evaluation of all the female partners by expert Gynaecologists, which included a detailed medical, reproductive and family history as well as a general and gynaecological physical examination. Furthermore, the ovulatory status, ovarian reserve testing, the structure and patency of the female reproductive tract were requested in all cases. The text has been revised accordingly.

We hope that the paper is now suitable to be considered for publication in the Original Articles section of PLOS ONE.

Sincerely yours,

Andrea Salonia, on behalf of all the authors

Andrea Salonia, MD, PhD, FECSM

Vita-Salute San Raffaele University

Division of Experimental Oncology/Unit of Urology, URI-Urological Research Institute

IRCCS Ospedale San Raffaele

Email: salonia.andrea@hsr.it

---

## [Editor Report · Decision Letter 3]

2 Jan 2023

Infertile men with semen parameters above WHO reference limits at first assessment may deserve a second semen analysis: challenging the guidelines in the real-life scenario

PONE-D-22-12621R3

Dear Dr. Salonia,

We’re pleased to inform you that your manuscript has been judged scientifically suitable for publication and will be formally accepted for publication once it meets all outstanding technical requirements.

Kind regards,

Stefan Schlatt

Academic Editor

PLOS ONE

Additional Editor Comments (optional):

The authors have responsed adequately and the paper is now acceptable for publication.
---

## [Editor Report · Acceptance letter]

8 Jan 2023

PONE-D-22-12621R3 

Infertile men with semen parameters above WHO reference limits at first assessment may deserve a second semen analysis: challenging the guidelines in the real-life scenario 

Dear Dr. Salonia:

I'm pleased to inform you that your manuscript has been deemed suitable for publication in PLOS ONE. Congratulations! Your manuscript is now with our production department. 

Kind regards, 

on behalf of

Dr. Stefan Schlatt 

Academic Editor

PLOS ONE